# Analysis of Physical Activity on Mental Hyperactivity, Sleep Quality, and Bodily Pain in Higher Education Students—A Structural Equation Model

**DOI:** 10.3390/healthcare12181841

**Published:** 2024-09-13

**Authors:** Rubén Fernández-García, Eduardo Melguizo-Ibáñez, José Manuel Hernández-Padilla, José Manuel Alonso-Vargas

**Affiliations:** 1Department of Nursing, Physiotherapy and Medicine, University of Almeria, 04120 Almeria, Spain; rubenfer@ual.es (R.F.-G.); j.hernandez-padilla@ual.es (J.M.H.-P.); 2Department of Didactics of Musical, Plastic and Corporal Expression, University of Granada, 18012 Granada, Spain

**Keywords:** physical activity, mental hyperactivity, body pain, sleep quality

## Abstract

The university population is at a vital stage of human development for acquiring an active lifestyle. Following this lifestyle will bring benefits in adulthood. This study seeks to analyse the relationship between physical activity and bodily pain, mental hyperactivity, and sleep quality as a function of the intensity of physical activity. A comparative, descriptive, and exploratory study is presented in a sample of 506 university students. The International Physical Activity Questionnaire, the Chronic Pain Grade Questionnaire, the Mental Hyperactivity Questionnaire, and the Pittsburgh Sleep Quality Index were used. The proposed model analyses the relationships of physical activity to bodily pain, mental hyperactivity, sleep disturbances, and time to fall asleep. The fit of the different adjustment indices is satisfactory (X^2^ = 0.47, df = 1, pl = 0.48, IFI = 0.99, CFI = 0.97; NFI = 0.91; RMSEA = 0.01). The relational analysis shows a positive relationship of physical activity to bodily pain (r = 0.02; *p* < 0.01), mental hyperactivity (r = 0.054; *p* < 0.01), sleep disturbances (r = 0.029; *p* < 0.01), and time to fall asleep (r = 0.047; *p* < 0.01). Multi-group structural equation analysis indicates that there are differences in the causal relationships of physical activity to sleep quality, mental hyperactivity, and bodily pain as a function of exercise intensity. The conclusion is that the intensity at which physical activity is performed plays a key role in mental and physical health.

## 1. Introduction

University students are facing a critical period in their lives [1]. The likelihood of unhealthy habits is very high [2]. This can have a negative impact on their health [2]. An increase in sedentary behaviours has been observed in this population [3]. People aged 18–64 years should follow several recommendations to obtain greater health benefits [4]. They should engage in more than 300 min of moderate-intensity aerobic exercise or 150 min of vigorous-intensity aerobic exercise [4]. Time spent in sedentary activities should be limited and replaced by light-intensity physical activity [4]. The intensity at which physical activity is performed brings different health benefits to individuals [5].

The level of sedentary lifestyles in the university population is associated with increased bodily pain [6]. Research points to the prevalence of back and neck pain [7]. This is mainly due to the high number of hours these students spend sitting [7]. These inadequate positions, sustained for a long time, are the cause of an aggressive load on the muscular and articular system [7]. This whole postural phenomenon not only brings alterations at the level of the musculoskeletal system [8]. An increase in the probability of suffering from diseases related to obesity, diabetes, or metabolic disorders has been observed [6]. Meta-analysis studies indicate that moderate-intensity physical activity reports fewer pain locations than vigorous physical activity [8]. Lower exercise intensity has been found to report higher pain locations [8].

Increased sedentary lifestyles in the university population are associated with mental disorders [9]. The default mode network has acquired a high degree of relevance in the study of cognitive processes [10]. When performing a perceptual or motor task that requires attentional processes, the activity of the default mode network decreases [10]. When an individual reflects on memories or personal experiences, this same network remains constantly activated [10]. Aerobic exercise can improve neurocognitive functions [11]. The acute effects of aerobic exercise may be associated with an increase in task-related cognitive arousal [11]. This results in cognitive function improvements in response to moderate-intensity exercise [11].

Bodily pain, a sedentary lifestyle, and cognitive activity are variables that condition the quality of students’ sleep [12,13,14]. Pain populations have been categorised as patients with sleep disorders [15]. Sleep issues, such as trouble falling asleep, insufficient sleep, and poor sleep quality, can result in various physical and mental health problems [13]. These sleep disturbances can elevate subjective pain intensity and exacerbate pain sensitisation in healthy individuals [16]. There is evidence that sleep problems are associated with various mental health difficulties [17]. Poor sleep has also been associated with post-traumatic stress [13], eating disorders [18], and psychosis spectrum experiences [13]. Engaging in regular physical activity can enhance both sleep quality and duration [19]. Other research indicates that physical activity can alleviate sleep disorders [19]. Additionally, moderate-intensity aerobic exercise has been shown to improve sleep in participants [19].

The originality of this research lies in the inclusion of all variables and in the multi-group structural equation analysis. There has been research analysing the causal relationships of physical activity on these variables separately [13,20,21]. There is no study using multi-group structural equation analysis to analyse the association between variables according to the level of intensity of physical activity. All these variables are related to the quality of life and rest in university students. It is necessary to understand the interaction between these variables through structural equation modelling. In addition, through the multi-group model, it is possible to analyse the influence of the intensity of physical activity on the variables that make up the model.

The objective of this research is:**O.1.** To analyse the relationship between physical activity and bodily pain, mental hyperactivity, and sleep quality as a function of the intensity of physical activity.

The research hypotheses are:

**H1.** 
*The intensity at which physical activity is performed will determine the relationship between physical activity and mental hyperactivity.*


**H2.** 
*High-intensity physical activity will be positively related to bodily pain.*


**H3.** 
*Mental hyperactivity will be positively associated with bodily pain regardless of the intensity of physical activity.*


**H4.** 
*Mental hyperactivity and bodily pain will be positively associated with sleep disturbances and time to fall asleep regardless of the intensity of physical activity.*


Based on these research hypotheses, the following theoretical model has been proposed (Figure 1).

## 2. Materials and Methods

### 2.1. Design and Participants

The steps outlined in the STROBE declaration were followed to carry out this research [22]. A cross-sectional, comparative, exploratory and descriptive study was carried out. The initial sample consisted of 541 participants. Thirty-five participants were eliminated for not responding adequately to the questions asked. The final research sample consisted of 506 university students (25.95 ± 2.76). Non-probability and convenience sampling was used. The students were from the University of Almeria and the University of Granada. The students belonged to the third and fourth year of the academic degrees of physiotherapy and physical activity and sport sciences. The gender distribution was as follows: 58.7% (n = 297) identified themselves as female and 41.3% (n = 209) identified themselves as male. The sampling error of this study was 0.033, with a confidence interval of 95.0%. The inclusion criteria consisted of being a university student and over 18 years of age. No exclusion criteria could affect the analyses.

### 2.2. Instruments and Variables

#### 2.2.1. Physical Activity

The International Physical Activity Questionnaire (Short-Form) provides information on the time and frequency dedicated to activities of varying intensities [23]. This consists of seven questions with acceptable measurement properties for monitoring physical activity levels in various settings. Three characteristics of physical activity are assessed: intensity (light, moderate, or vigorous), frequency (days per week), and duration (time per day) [24]. This instrument has previously been reliable and used in research with university students [25,26,27]. The Cronbach’s alpha value was α = 0.76. The McDonald’s omega value was ω = 0.78.

#### 2.2.2. Body Pain

The Chronic Pain Rating Questionnaire was used. The original version was not used [27]. The version adapted into Spanish [25] was used. Through eight items, the incidence of pain in the performance of different tasks is assessed. These items are rated on a 10-point Likert scale. An overall assessment of bodily pain is obtained by means of a mean value [28]. The Cronbach’s alpha showed a value of α = 0.82. The McDonald’s omega value was ω = 0.86.

#### 2.2.3. Mental Hyperactivity

The mental hyperactivity questionnaire [10] was used. This instrument was developed by Fernández-García et al. [10]. Through 10 items, it assesses the activation of the default neural network during the last three months. A mean value related to the overall mental hyperactivity of the individual is obtained through a summation. This questionnaire has only been validated in Spanish [10]. It has also been validated in a university population [10]. Adequate reliability values were obtained for this instrument (α = 0.89; ω = 0.89).

#### 2.2.4. Sleep Disturbances and Time to Fall Asleep

The Pittsburgh Sleep Quality Index [29] was used. The Spanish version was used for this study [30]. This instrument has been used with university students [31]. It assesses subjective sleep quality, daytime dysfunction, sleep duration, habitual sleep efficiency, sleep latency, habitual sleep efficiency, sleep disturbances, and medication use through 19 items. Adequate reliability values were obtained for this instrument (α = 0.78; ω = 0.80).

### 2.3. Procedure

Once the instruments were decided, they were entered into the Google Forms platform. The questionnaire was created and university students were invited to participate. The study was publicised through social networks, inviting young people who met the inclusion criteria to participate. This questionnaire was sent to the e-mails of the different students at the University of Granada and the University of Almeria. To ensure that the questionnaire was properly answered, three items were duplicated. In cases where the answers to these questions did not match, the subject’s participation was eliminated. Thirty-five responses were eliminated. Data were collected between January and March 2024. All participants participated voluntarily after giving informed consent. The ethical criteria set out in the Declaration of Helsinki have been followed. In addition, the study has been continuously monitored by the ethics committee of the University of Granada (2966/CEIH/2022).

### 2.4. Data Analysis

IBM SPSS 25.0 was used for the descriptive study and to build the multi-group structural equation model (IBM Corp., Armonk, NY, USA). A frequency study was carried out for the descriptive analysis. The study of the normality was carried out through the values of skewness and kurtosis of each item. The values of skewness and kurtosis should conform to the conventional criteria of normality. The value of skewness should be between −1.5 and 1.5. The kurtosis value should be between −3 and 3 [32,33]. The reliability of the instruments used was assessed by means of the Cronbach’s alpha and McDonald’s omega tests, with a reliability index of 95%. The level of correlation can be interpreted as a function of the value [34]. A value of r between 0 and 0.1 indicates no correlation [34]. A value between 0.10 and 0.29 shows a low correlation [34]. Values between 0.30 and 0.49 show a medium association [34]. A high correlation is between 0.50 and 0.69 [34]. Finally, a value between 0.70 and 1.0 indicates a very strong correlation [34].

Figure 1 presents the theoretical model that shows the direction of the associations. There is one variable that acts as an exogenous variable (physical activity), while the rest act as endogenous variables. This model analyses the relationships of physical activity to pain, mental hyperactivity, and sleep disturbances. The relationships of pain and mental hyperactivity to sleep disturbances and time to fall asleep are also analysed.

To fit the model, the values of the Incremental Fit Index (IFI) and Comparative Fit Index (CFI), together with the Normalised Fit Index (NFI), have been consulted. Their values must be greater than 0.90 to show a good fit [35]. The Root Mean Squared Error of Approximation (RMSEA) has also been consulted. The values of this index should be less than 0.08 [36].

## 3. Results

Table 1 shows the mean values, standard deviations, skewness, and kurtosis for each of the variables. It also presents the correlational matrix of the variables that make up the structural equation model. The correlational analysis shows positive and significant correlations between all the variables that make up the study. There is a positive correlation between physical activity and bodily pain (r = 0.02; *p* < 0.01). Similarly, bodily pain correlated positively with mental hyperactivity (r = 0.452; *p* < 0.01). A positive correlation was found between physical activity and mental hyperactivity (r = 0.054; *p* < 0.01). A positive and significant correlation was observed between sleep disturbances and physical activity (r = 0.29; *p* < 0.01), body pain (r = 0.430; *p* < 0.01), and mental hyperactivity (r = 0.494; *p* < 0.01). Finally, a positive correlation was found between the time it took to fall asleep and sleep disturbances (r = 0.382; *p* < 0.01), mental hyperactivity (r = 0.376; *p* < 0.01), body pain (r = 0.278; *p* < 0.01), and physical activity (r = 0.047; *p* < 0.01).

A confirmatory factor analysis was performed. The fit indices of the equation model evaluated were as follows: X^2^ = 0.47, df = 1, pl = 0.48, IFI = 0.99, CFI = 0.97; NFI = 0.91; and RMSEA = 0.01. Due to the good fit of the different indices consulted [35,36], it has been used as a model on which to develop the multi-group analysis. A confirmatory factor analysis was also carried out for the different instruments. The values for each scale are shown as follows: IPAQ-SF (IFI = 0.981, CFI = 0.980; NFI = 0.900; RMSEA = 0.073); Pittsburgh Sleep Quality Index (IFI = 0.911, CFI = 0.900; NFI = 0.884; RMSEA = 0.080); Mental Hyperactivity Questionnaire (IFI = 0.901, CFI = 0.920; NFI = 0.896; RMSEA = 0.039); and Chronic Pain Rating Questionnaire (IFI = 0.982, CFI = 0.900; NFI = 0.922; RMSEA = 0.084). These results meant that no dimensions were removed from the instruments.

Table 2 together with Figure 2, Figure 3 and Figure 4 present the standardised regression weights of the structural equation model. The relationship of physical activity to mental hyperactivity is positive for all three levels of physical activity intensity. Statistically significant differences are denoted for the three groups (*p* < 0.05). It is observed that moderate physical activity (β = 0.158) shows a stronger association with mental hyperactivity than light (β = 0.063) and vigorous physical activity (β = 0.049).

Continuing with the association of physical activity with bodily pain, vigorous physical activity shows a negative relationship (β = −0.027). In contrast, moderate (β = 0.069) and light physical activity (β = 0.112) have a positive relationship to bodily pain. Continuing with the relationship of physical activity to sleep disturbances, light (β = −0.177) and vigorous physical activity (β = −0.064) show a negative effect.

The association of mental hyperactivity with bodily pain has been shown to be statistically significant (*p* < 0.0001). Participants who engage in vigorous physical activity (β = 0.522) show a greater relationship than those who engage in light (β = 0.500) or moderate (β = 0.348) physical activity. A statistically significant relationship of mental hyperactivity to sleep disturbances is observed (*p* < 0.0001). Participants who engage in light physical activity (β = 0.420) show a greater association than those who engage in moderate (β = 0.353) or vigorous (β = 0.366) physical activity. A statistically significant relationship of mental hyperactivity to the time to fall asleep is also observed (*p* < 0.05). Students who engage in light physical activity (β = 0.279) show a greater relationship than those who engage in moderate (β = 0.185) or vigorous physical activity (β = 0.262).

A positive relationship of bodily pain to sleep disturbances is shown (*p* < 0.0001). Participants who engage in light physical activity (β = 0.428) are found to have a greater relationship than those who engage in vigorous (β = 0.237) or moderate (β = 0.202) physical activity. A significant relationship of bodily pain to the time to fall asleep is denoted (*p* < 0.0001). Young people who engage in moderate physical activity (β = 0.245) show a greater effect than those who engage in light (β = 0.191) or vigorous (β = 0.225) physical activity.

Finally, participants who engage in light physical activity show a greater relationship of sleep disturbances to the time to fall asleep (β = 0.165). Young people who engage in light physical activity (β = 0.165) show a greater relationship than those who engage in moderate physical activity (β = 0.030) of sleep disturbances to the time to fall asleep.

## 4. Discussion

The results indicate that physical activity correlates weakly with body pain, sleep disturbances, and time to fall asleep. In addition, the structural equation models show that the intensity at which physical activity is performed conditions the relationships between the variables.

The relationship of physical activity to mental hyperactivity is positive for all three levels of physical activity intensity. Moderate physical activity has a stronger impact on mental hyperactivity compared to light or vigorous physical activity.

The mechanisms through which physical activity influences cognition vary depending on the type and intensity of the activity [37]. The effects of different types of physical activity on brain activation have been compared [38]. Specific effects were obtained for different training programs [35]. A reduction in sensorimotor network activation was observed in the group receiving the aerobic intervention [38] An increase in activation in the visual–spatial network and in subcortical structures was also found [38]. The mechanisms by which physical activity affects cognition are intensity-dependent [37,38].

Continuing with the relationship of physical activity to bodily pain, vigorous activities show a negative association. In contrast, moderate and light physical activity have a positive relation to bodily pain.

Systematic review studies indicate that no significant association between physical activity and bodily pain has been found [8,39]. The relationship between the two variables is inverted U-shaped [36]. Other studies have found a linear association between the two variables [40]. Regarding the intensity of physical activity, activities with a higher intensity report higher contact with the contact surface [37]. This is related to the number of impacts with the ground [41]. If there has not previously been an educational process related to the impact zone, there may be a risk of pain [41]. If physical activity is supervised by a professional, it can be beneficial for joint strengthening [41].

The relationship of mental hyperactivity to bodily pain shows that participants who engage in vigorous physical activity show a greater association than those who engage in light or moderate physical activity.

Pain is fundamentally a construct that influences both physiological and psychological domains [42]. The perception of pain is partly reliant on cognitive processes such as learning, memory, and decision-making [42]. Various aspects of cognitive function indicate that higher levels of pain are linked to reduced cognitive abilities [43]. There is substantial evidence supporting the connection between physical inactivity and cognitive decline [43]. Engaging in physical activity may enhance brain vascular health by reducing blood pressure, optimizing the lipoprotein profile, and increasing cerebral blood flow [43].

Participants who engage in light physical activity show a greater relationship of bodily pain and mental hyperactivity to sleep disturbances.

Sleep quality plays a vital role in the preservation of health [44]. High-intensity physical activities have been found to report poor sleep quality [44]. Physical activities of light–moderate intensity have been shown to improve sleep quality by decreasing sleep latency [45]. Bodily pain has been reported to have a negative effect on sleep quality [45]. This is mainly due to the difficulty in falling asleep [45]. Practicing physical activity according to basic criteria can help to improve sleep quality [45]. The duration and intensity of exercise should be adapted to the type and location of the pain [45].

## 5. Strengths and Limitation

The reliance on self-reported questionnaires is a limitation. While validated instruments were used, self-reported data can be subject to bias. Including objective measures, such as accelerometers for activity levels, could enhance the accuracy of the findings [46,47]. It should be noted that non-probability and convenience sampling was used.

While the study shows the relationships between physical activity intensity and health outcomes, it does not explore the underlying physiological or psychological mechanisms in depth. Future research could focus on understanding why different intensities of physical activity influence these variables differently.

Although the authors acknowledge pain, sleep quality, and mental hyperactivity as interconnected, other confounding factors (such as diet, stress levels, or academic workload) might also play a role in the observed relationships but were not controlled for in the study. In terms of the strengths of the research, it should be noted that the data were collected using validated instruments. This makes the conclusions obtained reliable and valid. We have also worked with a large sample of university students from the south of Spain. It should be noted that this study provides a multifaceted view of the variables analysed. Based on these results, it would be possible to design a study focused on the application of physical exercise at different intensities on different psychosocial variables.

## 6. Conclusions

The conclusions of this research show that the intensity at which physical activity is performed conditions the relationship of the variables that make up the theoretical model. Physical activity at light intensity shows a greater effect of physical activity and mental hyperactivity on bodily pain. A better causal relationship of mental hyperactivity and bodily pain with sleep disturbances is also observed. In addition, this level of intensity also reports a better effect of mental hyperactivity and sleep disturbances on the time to fall asleep. As for the medium intensity, a better effect of physical activity on mental hyperactivity, and bodily pain on sleep duration, is observed. In view of these findings, it is necessary to take into account the intensity at which physical exercise is performed.

## Figures and Tables

**Figure 1 healthcare-12-01841-f001:**
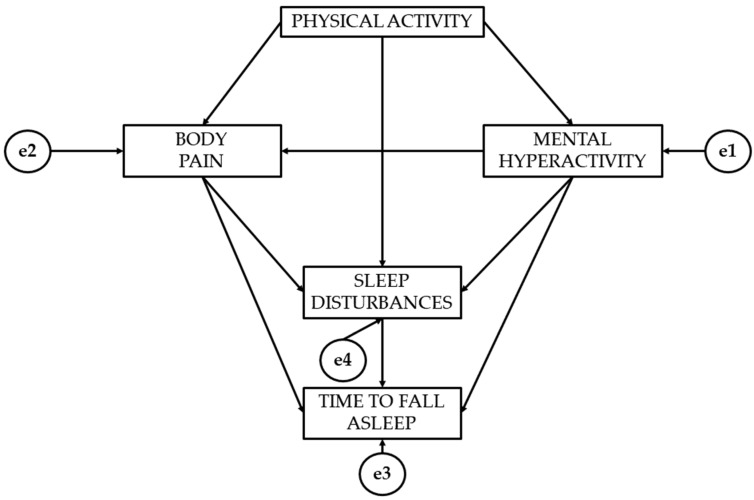
Theoretical research model.

**Figure 2 healthcare-12-01841-f002:**
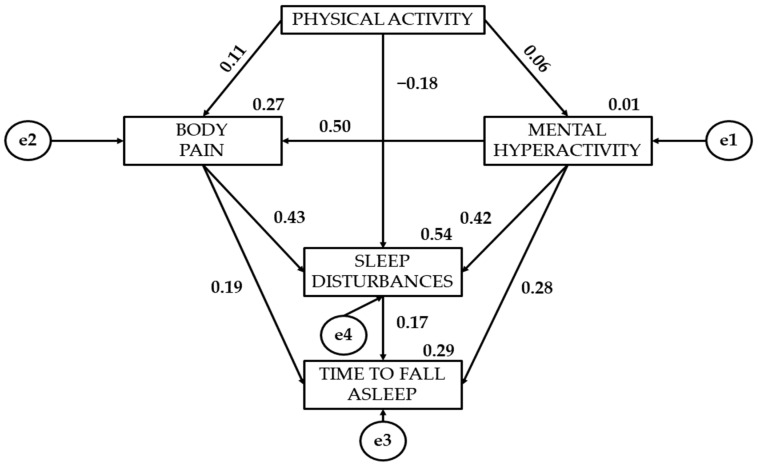
Regression weights for participants engaged in light physical activity.

**Figure 3 healthcare-12-01841-f003:**
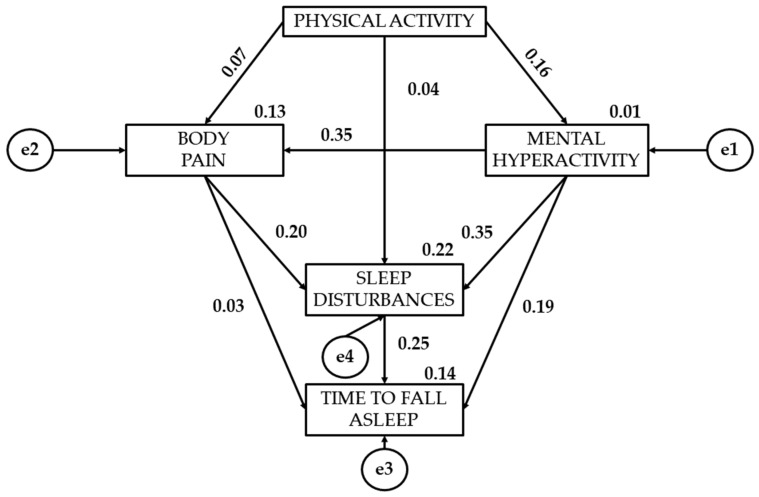
Regression weights for moderately active participants.

**Figure 4 healthcare-12-01841-f004:**
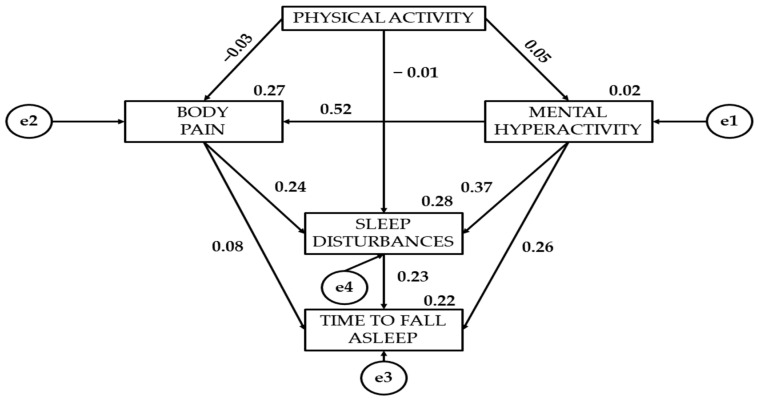
Regression weights for participants engaged in vigorous physical activity.

**Table 1 healthcare-12-01841-t001:** Descriptive, reliability, and correlational analyses of the study variables.

	M ± SD	Skew	Kur	2	3	4	5
1. Physical activity	4794.9852 ± 13,590.86792	0.934	0.156	0.02 *	0.054 *	0.029 *	0.047 *
2. Body pain	3.4769 ± 2.41065	0.450	−0.605	1	0.452 *	0.430 *	0.278 *
3. Mental hyperactivity	1.2132 ± 0.69496	0.390	−0.507		1	0.494 *	0.376 *
4. Sleep disturbances	0.7487 ± 0.42700	0.850	0.906			1	0.382 *
5. Time to fall asleep	1.8854 ± 0.88927	0.684	−0.424				1

Note: * *p* < 0.01.

**Table 2 healthcare-12-01841-t002:** Standardised regression weights for the multi-group equation model.

Effect Direction	Standardised Regression Weights			*p*
	Light	Moderate	Vigorous	
Mental hyperactivity ← physical activity	0.063	0.158	0.049	<0.05
Body pain ← physical activity	0.112	0.069	−0.027	0.248
Body pain ← mental hyperactivity	0.5	0.348	0.522	<0.0001
Sleep disturbances ← mental hyperactivity	0.42	0.353	0.366	<0.0001
Sleep disturbances ← body pain	0.428	0.202	0.237	<0.0001
Sleep disturbances ← physical activity	−0.177	0.041	−0.064	0.47
Time to fall asleep ← body pain	0.191	0.245	0.225	<0.0001
Time to fall asleep ← mental hyperactivity	0.279	0.185	0.262	<0.05
Time to fall asleep ← sleep disturbances	0.165	0.03	0.081	0.316

## Data Availability

The data used to support the findings of the current study are available from the corresponding author upon request.

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
