# Peer review of "Analysis of Physical Activity on Mental Hyperactivity, Sleep Quality, and Bodily Pain in Higher Education Students—A Structural Equation Model"

_healthcare, 2024, doi:10.3390/healthcare12181841_

Round 1

Reviewer 1 Report

Comments and Suggestions for Authors

Interesting study. I suggest changes before it is feasible to publish it.

A cross-sectional study does not allow establishing causal relationships. Although they do allow establishing associations, this is not possible if one variable is caused by another. For this reason I suggest eliminating “To analyse the causal relationships…” from the objectives and only saying that they seek to associate variables. Similar sentences should be searched for in the text and in discussions. Although they mention this limitation of this type of study in discussions, the word causal relationship is mentioned throughout the manuscript, but that is not adequate, according to its own limitations.

The questionnaire was created and university students were invited to participate. The study was publicised through social networks, inviting young people who met the inclusion criteria to participate: mentioning which universities, which courses, which cities. Also mentioning that the sampling was non-probabilistic and for convenience, also mentioning this in the study limitations.

This study was approved and supervised by an ethics committee (2966/CEIH/2022). Please mention which institution the ethics committee is from.

This model analyzes the causal relationship of physical activity on pain, mental hyperactivity and sleep disturbances: By definition, the causal relationship could only be established in longitudinal studies. Correct or justify why this model can be established in a cross-sectional study.

Results

Many of the correlations, although statistically significant, have a very low r, and it is probably a weak or slight correlation. It would be interesting to describe the degree of correlation, so that the reader knows that although they are related, the correlation was not so relevant.

I suggest adding in discussions a simple interpretation of the results linked to strategies that could be applied to university students to help solve the problems analyzed. This only with the strongest results from a statistical point of view. The above so that the study can be interpreted appropriately and can provide information for the generation of future intervention guides for university students.

Author Response

REVIEWER 1

Comment 1

A cross-sectional study does not allow establishing causal relationships. Although they do allow establishing associations, this is not possible if one variable is caused by another. For this reason I suggest eliminating “To analyse the causal relationships…” from the objectives and only saying that they seek to associate variables. Similar sentences should be searched for in the text and in discussions. Although they mention this limitation of this type of study in discussions, the word causal relationship is mentioned throughout the manuscript, but that is not adequate, according to its own limitations

Response 1

Thank you very much for your suggestion. The authors agree with you. We have proceeded to delete everything related to causal relationship.

Comment 2

The questionnaire was created and university students were invited to participate. The study was publicised through social networks, inviting young people who met the inclusion criteria to participate: mentioning which universities, which courses, which cities. Also mentioning that the sampling was non-probabilistic and for convenience, also mentioning this in the study limitations.

Response 2

Thank you very much for your comment. The requested information has been added

Comment 3

This study was approved and supervised by an ethics committee (2966/CEIH/2022). Please mention which institution the ethics committee is from.

Response 3

Thank you very much for your comment. The requested information has been added

Comment 4

This model analyzes the causal relationship of physical activity on pain, mental hyperactivity and sleep disturbances: By definition, the causal relationship could only be established in longitudinal studies. Correct or justify why this model can be established in a cross-sectional study.

Response 4

Thank you very much for your comment. The requested information has been corrected.

Comment 5

Many of the correlations, although statistically significant, have a very low r, and it is probably a weak or slight correlation. It would be interesting to describe the degree of correlation, so that the reader knows that although they are related, the correlation was not so relevant.

Response 5

Thank you very much for your suggestion. The values of r have been indicated.

Comment 6

I suggest adding in discussions a simple interpretation of the results linked to strategies that could be applied to university students to help solve the problems analyzed. This only with the strongest results from a statistical point of view. The above so that the study can be interpreted appropriately and can provide information for the generation of future intervention guides for university students.

Response 6

Thank you very much for your comment. A brief contextualisation of the results has been added at the beginning of the discussion.

Reviewer 2 Report

Comments and Suggestions for Authors

Estimados autores

The effort put into this research is appreciated, as it contributes to studying the health benefits of physical activity, especially in university students, a population particularly exposed to habits that can deteriorate their well-being. Below are some comments that I hope will contribute to improving your manuscript:

Introduction 

Although the introduction provides background supporting the connection between the variables studied and those proposed in the model, it needs to be clarified whether a real gap in knowledge is being addressed. That is, whether an explanatory model of this type has yet to be studied before is not specified. In this sense, it would be helpful to report similar research that has used explanatory models using structural equations or mediation models with some of the variables proposed in the present study to demonstrate that this is the first study that connects all these variables through an SEM model. On the other hand, it is also necessary for the authors to justify the study's relevance in the introduction; a helpful resource would be to mention the practical implications of the findings or results of this study. All of the above is vital to support the originality of your research, otherwise, there is no certainty that you are trying to report something that has yet to be studied.

Lines 65-68: The first two objectives seem to be sub-objectives of the third objective, which seems to be the general objective. I suggest eliminating the first two and leaving only the last one for a better understanding.

On the other hand, relationship or mediation hypotheses should be drafted based on the stated background to support the proposed theoretical model. Finally, Figure 1 should be placed at the end of the introduction, after the study hypotheses.

Material and Methods

Desing and participans

Line 74: It is advisable to mention whether any guidelines, such as STROBE (Strengthening the Reporting of Observational Studies in Epidemiology), were followed to guide the design of the study.

Line 75: More details about the characteristics of the sample, such as the context and location, are necessary. For example, specify whether the participants come from one or several universities, which courses of study, and from which locality and country. In addition, they must decide whether they will work with 1 or 2 decimals and maintain consistency throughout the text.

Line 77: "All participants participated voluntarily after giving informed consent." This has already been described in greater detail in the procedure section, so it should be removed and moved to that section.

Lines 77-79: It should be indicated whether there were exclusion criteria and what they were. If not, mention that no exclusion criteria could affect the analyses, such as health conditions that prevent physical activity or that affect sleep quality, such as pregnancy.

Instruments and variables

In this section, it is best to name the variable first and then the instrument with which it was measured, and then go ahead and describe the instrument. It is necessary to cite the original authors and describe the instrument in terms of the number of items, whether it is grouped into dimensions, the method for obtaining the scores, and whether it has been validated or used in the population studied; the latter should also be cited.

Procedure

Lines 101-102: This information is less relevant, and is common to all research, so mentioning it is unnecessary and should be eliminated. In the procedures section, it is important to indicate the periods in which the data were collected and how they were applied; in this case, it was through an online questionnaire using Google Forms. But through what method was this questionnaire distributed? The type of sampling should also be specified: was it probabilistic or not? by convenience? or voluntary? Was there any incentive for participation?

Lines 106-108: This information corresponds to an exclusion criterion and should be included in the participant's section. Therefore, from the 506 participants, 35 should be subtracted to obtain the final sample size. Please explain and detail the initial and final sample in the participants section.

Lines 109-110: "They were assured that the data would be processed for scientific purposes and anonymously." This information is inherent to all research that follows ethical standards; therefore, it is assumed that an ethics committee of an institution approves the research. The statement should be removed. This body and the ethical guidelines that were followed should be mentioned, such as the Declaration of Helsinki, which should be duly cited.

Data analysis

Lines 137-138: This information should be transferred to the results section. There, you should detail how this level of fit was achieved, specifying what type of factor analysis was performed (it is advisable to include at least one confirmatory factor analysis) to determine the adequacy of the items to the instruments before integrating them into the structural equation model. If modifications were made to any of the instruments, such as eliminating items, the fit indices should be reported for the models with all the items and the adjusted models without those items.

Discussion

Line 197: This information is unnecessary, please delete it. The purpose of the discussion is clear.

Lines 198-200: Provide more details about your findings regarding analyzing relationships through structural equation modeling. Including the hypotheses in the introduction will help better organize and word the ideas in this section.

Lines 201-207: Separate the contrast of the evidence with your findings in a second paragraph. Leave the first paragraph to highlight the main findings of your study.

Lines 238-239: Delete this descriptive information and replace it with a subheading "Strengths and Limitations." In this section, you mention some limitations of your research, which is appropriate. You should add that self-reported physical activity tends to overestimate the activity levels actually performed, so an objective measurement, ideally through accelerometers, is always preferable. This should be mentioned and supported with citations (abundant evidence supports this). Another limitation is the type of sampling, which, although the authors should clarify, can be deduced to have been a convenience sample, which implies a possible bias in the sample, probably composed of students interested in physical activity and health. On the other hand, authors should report the strengths of their research. In addition, it is recommended that the authors mention the practical implications of their results at this point and propose guidelines for future research, usually derived from the limitations detected.

Conclusions

The wording of the conclusion should be revised. It should not be limited to repeating the research findings but also offer practical implications of the results and indicate guidelines for future research.

Author Response

REVIEWER 2

Comment 1

Although the introduction provides background supporting the connection between the variables studied and those proposed in the model, it needs to be clarified whether a real gap in knowledge is being addressed. That is, whether an explanatory model of this type has yet to be studied before is not specified. In this sense, it would be helpful to report similar research that has used explanatory models using structural equations or mediation models with some of the variables proposed in the present study to demonstrate that this is the first study that connects all these variables through an SEM model. On the other hand, it is also necessary for the authors to justify the study's relevance in the introduction; a helpful resource would be to mention the practical implications of the findings or results of this study. All of the above is vital to support the originality of your research, otherwise, there is no certainty that you are trying to report something that has yet to be studied.

Response 1

Thank you very much for your comment. All the requested information has been added

Comment 2

Lines 65-68: The first two objectives seem to be sub-objectives of the third objective, which seems to be the general objective. I suggest eliminating the first two and leaving only the last one for a better understanding.

Response 2

Thank you very much for your comment. The two objectives have been removed

Comment 3

On the other hand, relationship or mediation hypotheses should be drafted based on the stated background to support the proposed theoretical model. Finally, Figure 1 should be placed at the end of the introduction, after the study hypotheses.

Response 3

Thank you very much for your comment. The requested changes have been implemented.

Comment 4

Line 74: It is advisable to mention whether any guidelines, such as STROBE (Strengthening the Reporting of Observational Studies in Epidemiology), were followed to guide the design of the study.

Response 4

Thank you very much for your comment. The requested information has been indicated.

Comment 5

Line 75: More details about the characteristics of the sample, such as the context and location, are necessary. For example, specify whether the participants come from one or several universities, which courses of study, and from which locality and country. In addition, they must decide whether they will work with 1 or 2 decimals and maintain consistency throughout the text.

Response 5

Thank you very much for your suggestion. The requested information has been added and the criteria have been unified.

Comment 6

Line 77: "All participants participated voluntarily after giving informed consent." This has already been described in greater detail in the procedure section, so it should be removed and moved to that section.

Response 6

Thank you very much for your comment. Your suggestion has been implemented.

Comment 7

Lines 77-79: It should be indicated whether there were exclusion criteria and what they were. If not, mention that no exclusion criteria could affect the analyses, such as health conditions that prevent physical activity or that affect sleep quality, such as pregnancy.

Response 7

Thank you very much for your suggestion. The requested information has been indicated

Comment 8

In this section, it is best to name the variable first and then the instrument with which it was measured, and then go ahead and describe the instrument. It is necessary to cite the original authors and describe the instrument in terms of the number of items, whether it is grouped into dimensions, the method for obtaining the scores, and whether it has been validated or used in the population studied; the latter should also be cited.

Response 8

Thank you very much for your comment. The requested information has been added

Comment 9

Lines 101-102: This information is less relevant, and is common to all research, so mentioning it is unnecessary and should be eliminated. In the procedures section, it is important to indicate the periods in which the data were collected and how they were applied; in this case, it was through an online questionnaire using Google Forms. But through what method was this questionnaire distributed? The type of sampling should also be specified: was it probabilistic or not? by convenience? or voluntary? Was there any incentive for participation?

Response 9

Thank you very much for your comment. The requested information has been added.

Comment 10

Lines 106-108: This information corresponds to an exclusion criterion and should be included in the participant's section. Therefore, from the 506 participants, 35 should be subtracted to obtain the final sample size. Please explain and detail the initial and final sample in the participants section.

Response 10

Thank you very much for your comment. Your changes have been implemented.

Comment 11

Lines 109-110: "They were assured that the data would be processed for scientific purposes and anonymously." This information is inherent to all research that follows ethical standards; therefore, it is assumed that an ethics committee of an institution approves the research. The statement should be removed. This body and the ethical guidelines that were followed should be mentioned, such as the Declaration of Helsinki, which should be duly cited.

Response 11

Thank you very much for your comments. The suggested changes have been implemented

Comment 12

Lines 137-138: This information should be transferred to the results section. There, you should detail how this level of fit was achieved, specifying what type of factor analysis was performed (it is advisable to include at least one confirmatory factor analysis) to determine the adequacy of the items to the instruments before integrating them into the structural equation model. If modifications were made to any of the instruments, such as eliminating items, the fit indices should be reported for the models with all the items and the adjusted models without those items.

Response 12

Thank you very much for your comment. A confirmatory factor analysis has been carried out for each instrument used.

Comment 13

Line 197: This information is unnecessary, please delete it. The purpose of the discussion is clear.

Response 13

Thank you very much for your comments. The suggested changes have been implemented

Comment 14

Lines 198-200: Provide more details about your findings regarding analyzing relationships through structural equation modeling. Including the hypotheses in the introduction will help better organize and word the ideas in this section

Response 14

Thank you very much for your comment. The suggested changes have been implemented.

Comment 15

Lines 201-207: Separate the contrast of the evidence with your findings in a second paragraph. Leave the first paragraph to highlight the main findings of your study.

Response 15

Thank you very much for your comment. The suggested changes have been implemented.

Comment 16

Lines 238-239: Delete this descriptive information and replace it with a subheading "Strengths and Limitations." In this section, you mention some limitations of your research, which is appropriate. You should add that self-reported physical activity tends to overestimate the activity levels actually performed, so an objective measurement, ideally through accelerometers, is always preferable. This should be mentioned and supported with citations (abundant evidence supports this). Another limitation is the type of sampling, which, although the authors should clarify, can be deduced to have been a convenience sample, which implies a possible bias in the sample, probably composed of students interested in physical activity and health. On the other hand, authors should report the strengths of their research. In addition, it is recommended that the authors mention the practical implications of their results at this point and propose guidelines for future research, usually derived from the limitations detected.

Response 16

Thank you very much for your comment. The requested information has been added

Comment 17

The wording of the conclusion should be revised. It should not be limited to repeating the research findings but also offer practical implications of the results and indicate guidelines for future research.

Response 17

Thank you very much for your comment. Your comment has been implemented

Reviewer 3 Report

Comments and Suggestions for Authors

The topic of the paper is interesting and fits the scope of the journal. The text is relatively well written and composed. However, I have some comments that I believe would enhance this article.

Introduction

Page 1, lines 28-29. In these lines, the authors refer to both young individuals and older adults. In my opinion, they should focus solely on young individuals, such as those aged 18-22 years old.

Page 1, lines 41-42. Could you please clarify with whom the moderate intensity is being compared?

Page 1, lines 49-52. Can only aerobic exercise improve neurocognitive functions, or can strength training and other forms of exercise also contribute?

Methods

Page 2, line 75. Please use only two decimal places for the standard deviation (SD). For example, 2.76.

Limitation of this study

The reliance on self-reported questionnaires is a limitation. While validated instruments were used, self-reported data can be subject to bias (e.g., over-reporting or under-reporting physical activity). Including objective measures, such as accelerometers for activity levels, could enhance the accuracy of the findings.

While the study shows relationships between physical activity intensity and health outcomes, it does not explore the underlying physiological or psychological mechanisms in depth. Future research could focus on understanding why different intensities of physical activity influence these variables differently.

Although the authors acknowledge pain, sleep quality, and mental hyperactivity as interconnected, other confounding factors (such as diet, stress levels, or academic workload) might also play a role in the observed relationships but were not controlled for in the study.

Author Response

REVIEWER 3

Comment 1

Page 1, lines 28-29. In these lines, the authors refer to both young individuals and older adults. In my opinion, they should focus solely on young individuals, such as those aged 18-22 years old.

Response 1

Thank you very much for your comment.

In this case reference is made to the recommendations of the World Health Organisation. These include recommendations for all adults, i.e. people between 18 and 64 years of age. In response to your suggestion, the authors searched for physical activity recommendations from other organisations for the study population, but no recommendations were found.

Comment 2

Page 1, lines 41-42. Could you please clarify with whom the moderate intensity is being compared?

Response 2

Thank you very much for your comment. The authors have reworded the sentence as indicated.

Comment 3

Page 1, lines 49-52. Can only aerobic exercise improve neurocognitive functions, or can strength training and other forms of exercise also contribute?

Response 3

Thank you very much for your comment. Anaerobic exercise is more effective than anaerobic exercise in improving neurocognitive functions. In addition, aerobic exercise at moderate intensity is more effective at moderate intensity than at light or vigorous intensity in improving neurocognitive functions.

Comment 4

Page 2, line 75. Please use only two decimal places for the standard deviation (SD). For example, 2.76

Response 4

Thank you very much for your comment. Your suggestion has been implemented

Comment 5

The reliance on self-reported questionnaires is a limitation. While validated instruments were used, self-reported data can be subject to bias (e.g., over-reporting or under-reporting physical activity). Including objective measures, such as accelerometers for activity levels, could enhance the accuracy of the findings.

 While the study shows relationships between physical activity intensity and health outcomes, it does not explore the underlying physiological or psychological mechanisms in depth. Future research could focus on understanding why different intensities of physical activity influence these variables differently.

 Although the authors acknowledge pain, sleep quality, and mental hyperactivity as interconnected, other confounding factors (such as diet, stress levels, or academic workload) might also play a role in the observed relationships but were not controlled for in the study.

Response 5

Thank you very much for your comments. The authors have proceeded to add them to the research.

Round 2

Reviewer 1 Report

Comments and Suggestions for Authors

The authors have improved the manuscript and made changes based on all comments. Therefore, I suggest the publication of the manuscript.

Author Response

Dear Reviewer. 
Your suggestions have been satisfactorily implemented by the authors. We would like to thank you for your considerations to improve our work. 

Reviewer 2 Report

Comments and Suggestions for Authors

Dear authors

I appreciate the effort, the improvements in your manuscript are notable. For my part, I don't have any more comments; I only think that the summary should be improved. You must provide more specific information on the results of the structural model analyzed in a descriptive and statistical manner.

Author Response

Thank you very much for your comments for improvement. The authors have pointed out data in the summary. 

Reviewer 3 Report

Comments and Suggestions for Authors

Accept in present form.

Author Response

(The authors gave the same response as above.)
